# Phase-dependent trends in the prevalence of myalgic encephalomyelitis / chronic fatigue syndrome (ME/CFS) related to long COVID: A criteria-based retrospective study in Japan

**Satoru Morita, Kazuki Tokumasu, Yuki Otsuka, Hiroyuki Honda, Yasuhiro Nakano, Naruhiko Sunada, Yasue Sakurada, Yui Matsuda, Yoshiaki Soejima, Keigo Ueda, Fumio Otsuka** *

Department of General Medicine, Okayama University Graduate School of Medicine, Dentistry and Pharmaceutical Sciences, Okayama, Japan

* fumiotsu@md.okayama-u.ac.jp

## Abstract

**Data Availability Statement:** Detailed data will be available if requested to the corresponding author.

### Background

The characteristics of myalgic encephalomyelitis/chronic fatigue syndrome (ME/CFS) related to COVID-19 have remained uncertain. To elucidate the clinical trend of ME/CFS induced by long COVID, we examined data for patients who visited our outpatient clinic established in a university hospital during the period from Feb 2021 to July 2023.

### Methods

Long COVID patients were classified into two groups, an ME/CFS group and a non-ME/CFS group, based on three diagnostic criteria.

### Results

The prevalence of ME/CFS in the long COVID patients was 8.4% (62 of 739 cases; female: 51.6%) and factors related to ME/CFS were severe illness, smoking and alcohol drinking habits, and fewer vaccinations. The frequency of ME/CFS decreased from 23.9% in the Preceding period to 13.7% in the Delta-dominant period and to 3.3% in the Omicron-dominant period. Fatigue and headache were commonly frequent complaints in the ME/CFS group, and the frequency of poor concentration in the ME/CFS group was higher in the Omicron period. Serum ferritin levels were significantly higher in female patients in the ME/CFS group infected in the Preceding period. In the ME/CFS group, the proportion of patients complaining of brain fog significantly increased from 22.2% in the Preceding period to 47.9% in the Delta period and to 81.3% in the Omicron period. The percentage of patients who had received vaccination was lower in the ME/CFS group than the non-ME/CFS group over the study period, whereas there were no differences in the vaccination rate between the groups in each period.

However, public sharing of the data set has been restricted by the Ethics Committee of Okayama University Hospital, since public deposition would breach compliance with the protocol approved by our research ethics board. There are ethical restrictions on sharing the anonymized dataset. Because the analysis is based on the data from patients who were referred to and treated at our hospital and contains information that may identify the patients, this study is permitted under the restrictions of the research ethics committee of the institution. Furthermore, secondary use of the data is not permitted in this study by the Ethics Committee of Okayama University Hospital. Contact information for ethics committees other than the authors to whom data requests may be sent to the research promotion section of Okayama University Hospital (mae6605@adm.okayama-u.ac.jp).

**Funding:** This research was supported by AMED under grant number 22fk0108517h0001 and 23fk0108585h0001. The funders had no role in study design, data collection and analysis, decision to publish, or preparation of the manuscript.

**Competing interests:** The authors have declared that no competing interests exist.

**Abbreviations:** BBB, Blood-brain barrier; BMI, body mass index; CAC, COVID-19 aftercare clinic; CNS, central nervous system; COVID-19, coronavirus disease 2019; FAS, Fatigue Assessment Scale; IOM, Institute of Medicine; ME/CFS, myalgic encephalomyelitis/chronic fatigue syndrome; SDS, Self-Rating Depression Scale.

## Conclusion

The proportion of long COVID patients who developed ME/CFS strictly diagnosed by three criteria was lower among patients infected in the Omicron phase than among patients infected in the other phases, while the proportion of patients with brain fog inversely increased. Attention should be paid to the variant-dependent trends of ME/CFS triggered by long COVID (300 words).

## Introduction

It has been over four years since the novel coronavirus disease 2019 (COVID-19) was first discovered. Initially, COVID-19 was feared for its high degree of aggravation and high mortality rate in the acute phase. However, it gradually became apparent that some patients with COVID-19 suffered prolonged sequelae even after recovering from the acute stage. These sequelae are known as "long COVID" or "post COVID-19 condition" [1, 2], and it has been reported worldwide that long COVID presents a variety of symptoms including general fatigue, dysosmia, dysgeusia, headache, sleep disorder, and brain fog [3, 4].

Some long COVID patients have symptoms similar to those of myalgic encephalomyelitis/chronic fatigue syndrome (ME/CFS) [5, 6], and it has been reported that a certain proportion of the patients with long COVID develop ME/CFS [7–9]. ME/CFS is a degenerative disease characterized by various multisystemic symptoms, and ME/CFS is diagnosed on the basis of symptoms persisting for more than 6 months including pathological fatigue, which refers to significant physical and mental exhaustion that markedly decreases activity levels, is not a result of exertion, is not alleviated by rest and is unexplained, post-exertional fatigue, sleep disturbances, pain, neurological dysfunction, and cognitive disorder [10, 11]. Except for the finding indicating that serum ferritin level is a possible candidate for the development of ME/CFS related to long COVID [12], there has been no specific biomarker for ME/CFS, and there are therefore about 20 sets of criteria for ME/CFS including the Fukuda Criteria [10], the Canadian Consensus Criteria [11, 13], and the Institute of Medicine (IOM) Criteria [14, 15].

The prevalence of ME/CFS was estimated to be 0.42% in adult patients in a study conducted in the United States [16] and it was estimated to be 0.89% in studies involving 13 countries [14]. In addition, the proportion of long COVID patients meeting the IOM criteria was estimated to be approximately 10% [17]. Conversely, there have been reports indicating that up to almost half of patients with long COVID meet the criteria for ME/CFS [16, 18] Various questionnaire-based surveys were used in those studies to estimate the prevalence of ME/CFS following COVID-19 [14, 16, 17]. However, it is difficult to exclude the possibility of basal diseases presenting ME/CFS-like symptoms such as endocrine and/or autoimmune disorders in such questionnaire-based surveys. Moreover, it has been reported that the prevalence of ME/CFS varies significantly depending on the diagnostic criteria applied due to the difference in the clinical characteristic each criterion emphasizes [14]. Therefore, we have utilized three international sets of criteria for ME/CFS to standardize the clinical picture of ME/CFS in long COVID patients, and we showed in our earlier study that the prevalence of ME/CFS in long COVID patients was 16.8% [9].

Considering that there are many similarities in the underlying biology between long COVID and ME/CFS, it would be beneficial to investigate ME/CFS associated with long COVID [5, 6] not only for long COVID patients but also for ME/CFS patients. The aim of the present study was to elucidate the prevalence rate and the clinical characteristics of ME/CFS that developed from long COVID for each viral variant.

## Patients and methods

### Enrollment of long COVID patients

This study was performed in a single institution as a retrospective study. We established our COVID-19 aftercare clinic (CAC) on February 15, 2021 in the Department of General Medicine, Okayama University Hospital (Japan) for managing and evaluating patients who have suffered from post COVID-19 condition symptoms. Most of the patients who consulted the CAC were referred from outside medical facilities. The onset of COVID-19 in the patients was epidemiologically divided on the basis of reports of Okayama Prefecture in Japan into the Preceding period and the Delta- or Omicron-dominant period [19]. The Preceding phase includes the time period from the conventional strain to the Alpha-variant phase (the period before July 18, 2021), the Delta phase is when the Delta variants were dominant (the period from July 19, 2021 to December 31, 2021), and the Omicron phase is when the Omicron variants were dominant (the period after January 1, 2022) [19].

### Collection of clinical data related to ME/CFS

We retrospectively obtained clinical information for patients who visited our CAC. Medical records of 748 patients who visited our CAC from February 15, 2021 to July 27, 2023 were carefully reviewed between July 27, 2023 and November 30, 2023. We accessed medical records that included information enabling identification of individual participants and then the collected data were anonymized before analysis. We defined long COVID as a condition in which symptoms remain for longer than one month after the onset of COVID-19 [2]. Patients' information regarding age, sex, body mass index (BMI), smoking and alcohol-drinking habits, clinical severities in the acute phase of COVID-19 [20], various symptoms related to long COVID, vaccination histories, and time periods from the acute infection was obtained from medical records. We have performed face-to-face medical examinations for all patients who visited the CAC and investigated the possibilities of various diseases presenting symptoms similar to those of ME/CFS that are listed as conditions to be excluded according to the Canadian Consensus Criteria [11]. Blood samples were taken in a sitting position at the time when each patient visited the CAC. Assays for serum ferritin concentration were performed by an electrochemiluminescence immunoassay (ECLIA) using the Elecsys Ferritin kit (F. Hoffmann-La Roche AG, Basel, Switzerland).

We also evaluated the Fatigue Assessment Scale (FAS) [21] and Self-Rating Depression Scale (SDS) [22] by using questionnaires at the first visit. "Brain fog" symptoms were carefully examined on the basis of previous reports [23, 24] by individual interviews on the basis of complaints about the subjective feeling of concentration difficulty and defective ability to concentrate; for example, being mentally sluggish, spaced-out and fuzzy affecting the patient's ability and/or capacity to think or concentrate with an obtuse headache. The Fukuda Criteria [10], Canadian Consensus Criteria [11, 13] and IOM Criteria [25] are frequently used sets of criteria for diagnosing ME/CFS [9], and thus long COVID patients who met all of the three sets of criteria were diagnosed with ME/CFS to establish the ME/CFS group in the present study.

### Statistical analyses

We used Stata/SE 18.0 (StataCorp, 4905 Lakeway Dr, College Station, TX 77845, USA) for all statistical analyses. The data were statistically analyzed by using the Mann–Whitney U test for continuous variables and Pearson's $\chi^2$ test for categorical variables. A $p$ value of less than 0.05 was considered statistically significant.

**Table 1. Numbers and proportions of long COVID patients who met the ME/CFS criteria.**

| Types of Criteria | Number (%) of patients (n = 739) |
|---|---|
| Fukuda, 1994 | 106 (14.3) |
| Canada, 2003 | 73 (9.9) |
| IOM, 2015 | 116 (15.7) |
| All of the above criteria | 62 (8.4) |

## Ethical approval

The present study was approved (No. 2105–030) by the Ethics Committee of Okayama University Hospital and adhered to the Helsinki Declaration. In this study, all of the information for patients was obtained retrospectively and all of the information was anonymized. No prospective data were obtained and no new interventions were implemented. Therefore, we did not need obtain informed consent from patients. Information on this study was published on the hospital website and patients themselves were given the opportunity to not have personal data used for the present study. This is an ethical consideration, and the Ethics Board has approved this study.

## Results

We enrolled 748 patients in the present study, and we obtained data for 739 patients after excluding 5 patients who visited our CAC in less than 4 weeks after the onset of COVID-19 and 4 patients under the age of 10 years. There were no patients who obviously had a different disease other than ME/CFS such as endocrine, metabolic and autoimmune disorders. The percentage of patients who met all of the three sets of criteria, including the Fukuda Criteria, Canadian Consensus Criteria, and IOM Criteria, was 8.4% (Table 1). As shown in Fig 1, the

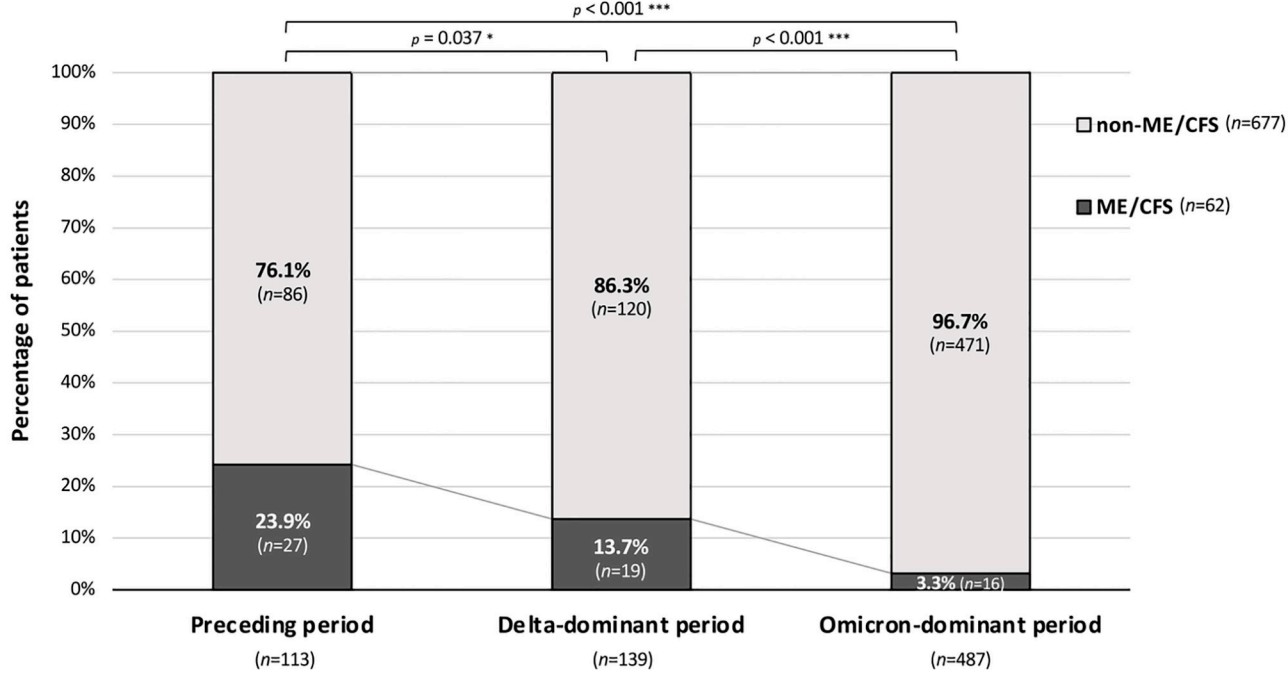

**Fig 1. Proportions of patients with ME/CFS related to long COVID who were infected during the three variant periods.** The percentages of ME/CFS patients and non-ME/CFS patients (ME/CFS: n = 62; non-ME/CFS: n = 677) are shown by the period in which they were infected. The $\chi^2$ test was performed for the proportions of ME/CFS patients between the variant phases, and *$p<0.05$ and ***$p<0.001$ were considered statistically significant.

**Table 2. Numbers and proportions of long COVID patients who met the ME/CFS criteria in each period.**

| Types of Criteria | Number (%) of patients in each period | | |
|---|---|---|---|
| | Preceding period (n = 113) | Delta period (n = 139) | Omicron period (n = 487) |
| Fukuda, 1994 | 27 (23.9) | 21 (15.1) | 58 (11.9) |
| Canada, 2003 | 29 (25.7) | 21 (15.1) | 23 (4.7) |
| IOM, 2015 | 30 (26.1) | 22 (15.8) | 64 (13.1) |
| All of the above criteria | 27 (23.9) | 19 (13.7) | 16 (3.3) |

phase-dependent proportion of patients with ME/CFS related to long COVID significantly decreased from 23.9% to 13.7% ($p<0.05$) and to 3.3% ($p<0.001$) in the Preceding period, Delta-dominant period and Omicron-dominant period, respectively. The proportions of patients who met the Fukuda Criteria were 23.9% during the Preceding period, 15.1% during the Delta-dominant period, and 11.9% during the Omicron-dominant period. The proportions of patients who met the Canadian Consensus Criteria in those three periods were 25.7%, 15.1% and 4.7%, respectively, and the proportions of patients who met the IOM Criteria in those periods were 26.1%, 15.8%, and 13.1%, respectively (Table 2).

A comparison of the clinical backgrounds of ME/CFS patients and those of non-ME/CFS patients is shown in Table 3. The median ages of patients in the ME/CFS and non-ME/CFS groups were 39 years and 41 years, respectively, without an apparent bias in the age distribution, in which 40 to 60 years of age were predominant in both groups (41.9% and 43.7%, respectively). The gender ratios were not significantly different but were slightly female dominant, with 51.6% of females in the ME/CFS group and 54.7% of females in the non-ME/CFS group. There were no significant differences between the two groups in BMI, administration of steroids, and duration from the onset of COVID-19 to the first CAC visit. The ME/CFS group had significantly higher rates of smoking habit (43.5% vs. 30.9%; $p<0.05$), alcohol drinking habit (50% vs. 32.9%; $p<0.01$), and admission in the acute phase of COVID-19 (33.9% vs. 13.0%; $p<0.001$) than those in the non-ME/CFS group. Nonetheless, because of the retrospective nature of this study, it was not clarified whether the patients actually had those habits before or after the onset of COVID-19. The severity of COVID-19 in the acute phase was defined by the Ministry of Health, Labour and Welfare in Japan [20]. The percentage of moderate to severe cases was significantly higher in the ME/CFS group than in the non-ME/CFS group (30.6% vs. 9.5%; $p<0.001$). Regarding vaccination status, the percentages of patients who received at least 1 dose and patients with no vaccination (48.4% and 50.0%, respectively) in the ME/CFS group were significantly different ($p<0.01$) from those in the non-ME/CFS group (66.8% and 32.2%, respectively). Regarding the duration from infection to the first CAC visit, periods within 3 months were predominant in both the ME/CFS and non-ME/CFS groups (45.2% vs 46.5%, respectively).

Vital signs including blood pressure, pulse rate, oxygen saturation, respiratory rate, and body temperature at the first visit to the CAC were not conspicuously different between the ME/CFS and non-ME/CFS groups as shown in Table 4. On the other hand, responses to questions in the questionnaires for detecting fatigue and depression levels were significantly worse in the ME/CFS group than in the non-ME/CFS group. Namely, the levels of FAS ($p<0.001$), including FAS physical ($p<0.001$) and FAS mental ($p<0.001$), and the levels of SDS ($p = 0.0013$) were significantly higher in ME/CFS patients than in non-ME/CFS patients (Table 4).

Regarding the clinical characteristics of ME/CFS related to long COVID, comparisons of the frequencies of symptoms between ME/CFS patients and non-ME/CFS patients are shown

**Table 3. Backgrounds of patients with ME/CFS related to long COVID and non-ME/CFS patients.**

|  | ME/CFS | non-ME/CFS | p-value |
|---|---|---|---|
|  | (n = 62) | (n = 677) |  |
| **Age (years):** |  |  |  |
| Total age, median [IQR] | 39 [28–50] | 41 [25–51] | 0.8401# |
| < 19 years, n (%) | 7 (11.3%) | 97 (14.3%) |  |
| 20–40 years, n (%) | 24 (38.7%) | 219 (32.3%) |  |
| 40–60 years, n (%) | 26 (41.9%) | 296 (43.7%) |  |
| > 60 years, n (%) | 5 (8.1%) | 65 (9.6%) |  |
| **Gender:** |  |  | 0.646 |
| Male, n (%) | 30 (48.4%) | 307 (45.3%) |  |
| Female, n (%) | 32 (51.6%) | 370 (54.7%) |  |
| **BMI:** |  |  |  |
| Total BMI, median [IQR] | 23.4 [20.9–27.1] | 22.4 [20.2–25.8] | 0.1948# |
| < 25, n (%) | 37 (59.7%) | 474 (71.0%) |  |
| 25–30, n (%) | 21 (33.9%) | 126 (18.9%) |  |
| > 30, n (%) | 4 (6.4%) | 68 (10.2%) |  |
| **Habits:** |  |  |  |
| Smoking, n (%) | 27 (43.5%) | 207 (30.9%) | 0.041* |
| Alcohol drinking, n (%) | 31 (50%) | 220 (32.9%) | 0.007* |
| **Acute phase status:** |  |  |  |
| Admission, n (%) | 21 (33.9%) | 88 (13.0%) | <0.001* |
| Use of steroids, n (%) | 9 (1.3%) | 57 (8.4%) | 0.098 |
| **Severity of COVID-19 in acute phase:** |  |  | <0.001* |
| Mild, n (%) | 43 (69.4%) | 612 (90.5%) |  |
| Moderate / Severe, n (%) | 19 (30.6%) | 64 (9.5%) |  |
| **Vaccination status of COVID-19:** |  |  | 0.003* |
| 0 dose, n (%) | 31 (50.0%) | 218 (32.2%) |  |
| At least 1 dose, n (%) | 30 (48.4%) | 452 (66.8%) |  |
| **Period from onset until the first visit:** |  |  |  |
| Median duration [IQR] | 103 [64–178] | 96 [62–145] | 0.3148# |
| < 3 months, n (%) | 28 (45.2%) | 314 (46.5%) |  |
| 3–6 months, n (%) | 19 (30.6%) | 243 (36.0%) |  |
| > 6 months, n (%) | 15 (24.2%) | 118 (17.5%) |  |

IQR: interquartile ranges. BMI: body mass index. The χ2 test and the Mann–Whitney U test# were performed for each symptom between the groups. We regarded *p<0.05 as statistically significant differences between the two groups.

in Fig 2. ME/CFS patients had higher frequencies of fatigue (p<0.001), insomnia (p<0.05) and dizziness (p<0.001) in the Preceding period, higher frequencies of fatigue (p<0.001), headache (p<0.001), dizziness (p<0.05) and chest pain (p = 0.008) in the Delta-dominant period, and higher frequencies of fatigue (p<0.01), headache (p<0.001), and poor concentration (p<0.01) in the Omicron-dominant period than those in non-ME/CFS patients (Fig 2). As shown in Fig 2, fatigue and headache were common major symptoms of ME/CFS related to COVID-19 throughout the variant periods.

There has been no specific parameter for detecting ME/CFS accompanied by long COVID. However, serum ferritin levels are known to reflect inflammatory responses and can be used to diagnose and estimate the severity of COVID-19 in female long COVID patients [12, 26]. Of

**Table 4. Physical and mental parameters for characterization of patients with ME/CFS related to long COVID and non-ME/CFS patients.**

| Clinical parameters | ME/CFS | (number) | non-ME/CFS | (number) | *p*-value |
|---|---|---|---|---|---|
| **SBP** (mmHg) | 129 [112–138] | (62) | 122 [111–136] | (668) | 0.1849 |
| **DBP** (mmHg) | 73 [63–81] | (62) | 72 [65–82] | (668) | 0.8998 |
| **PR** (bpm) | 82 [76–88] | (62) | 81 [73–90] | (666) | 0.7404 |
| **SpO2** (room air; %) | 98 [98–99] | (61) | 98 [98–99] | (664) | 0.3378 |
| **RR** (/min) | 18 [16–20] | (60) | 17 [14–20] | (660) | 0.3622 |
| **BT** (°C) | 36.8 [36.5–37.1] | (61) | 36.7 [36.5–37.0] | (663) | 0.1022 |
| **FAS** | 40 [33–43] | (62) | 32 [23–40] | (663) | <0.001* |
| **FAS physical** | 18 [16–19] | (62) | 15 [11–18] | (670) | <0.001* |
| **FAS mental** | 22 [18–25] | (62) | 17 [12–22] | (669) | <0.001* |
| **SDS** | 51 [46–60] | (62) | 48 [41–54] | (659) | 0.0013* |

Medians [IQR: interquartile ranges] are shown. SBP: systolic blood pressure, DBP: diastolic blood pressure, PR: pulse rate, SpO2: saturation of percutaneous oxygen, RR: respiratory rate, BT: body temperature, FAS: fatigue assessment scale, SDS: Self-rating Depression Scale. The Mann–Whitney U test was performed to compare the levels between the two groups. We regarded *$p<0.05$ as statistically significant differences between the two groups.

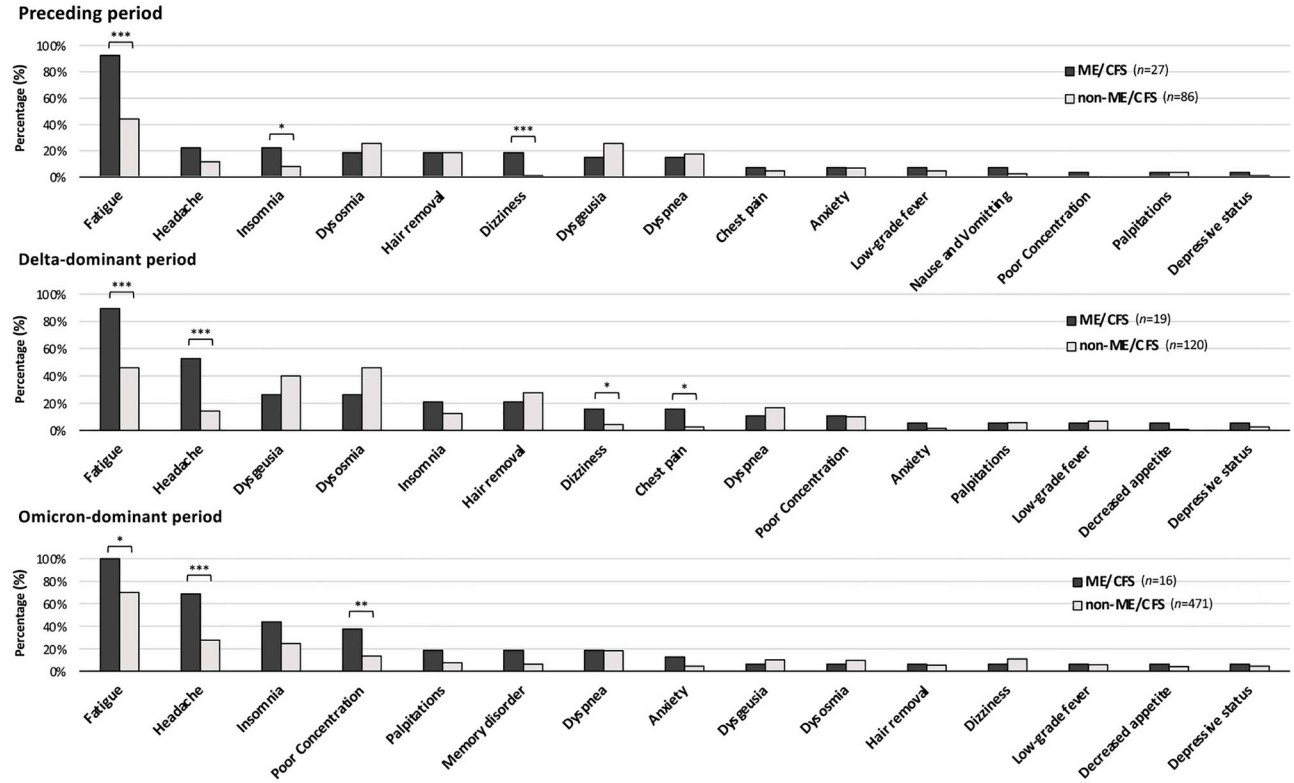

**Fig 2. Comparison of the clinical characteristics of long COVID patients in the ME/CFS group and the non-ME/CFS group in the three variant periods.** The percentages of patients with long COVID symptoms in the ME/CFS group (Total: n = 62; Preceding period: n = 27; Delta-dominant period: n = 19; Omicron-dominant period: n = 16) and non-ME/CFS group (Total: n = 677; Preceding period: n = 86; Delta-dominant period: n = 120; Omicron-dominant period: n = 471) are shown. The $\chi^2$ test was performed for each symptom between the two groups, and *$p<0.05$, **$p<0.01$, and ***$p<0.001$ were considered statistically significant.

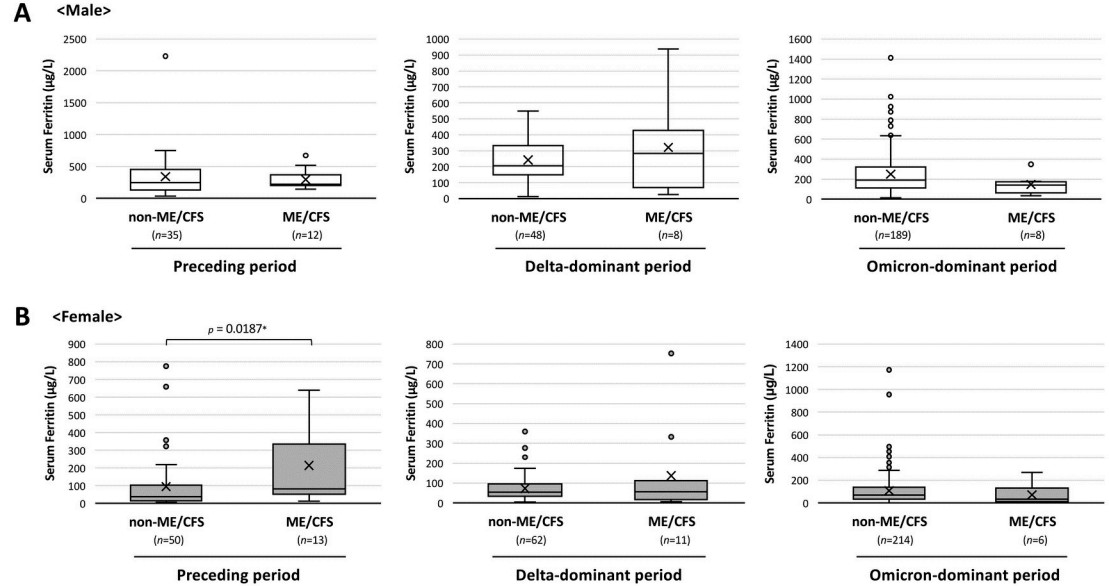

**Fig 3. Serum ferritin levels in long COVID patients in the ME/CFS group and non-ME/CFS group.** Serum ferritin levels are presented by sex (**A**: Male: n = 300; **B**: Female: n = 356) and separated by the onset periods of COVID-19 (Preceding period: n = 110; Delta-dominant period: n = 129; Omicron-dominant period: n = 417). The Mann-Whitney U test was performed to compare the levels between the two groups, and *$p$< 0.05 was considered statistically significant.

interest, a significant difference in serum ferritin levels between female ME/CFS patients and female non-ME/CFS patients ($p$<0.05) was found in the Preceding period but not in the Delta and Omicron phases (Fig 3).

Poor concentration and decreased memory are symptoms of "brain fog" [23, 24]. Long COVID patients have been shown to have various symptoms and to experience brain fog. As shown in Fig 4, patients in the ME/CFS group had a generally higher prevalence of brain fog than that in patients in the non-ME/CFS group, and the difference was significantly enhanced in the Delta-dominant period (18.3% vs. 47.4%; $p$<0.01) and Omicron-dominant period (30.2% vs. 81.3%; $p$<0.001) compared with that in the Preceding period (11.6% vs. 22.2%; $p$ = 0.168).

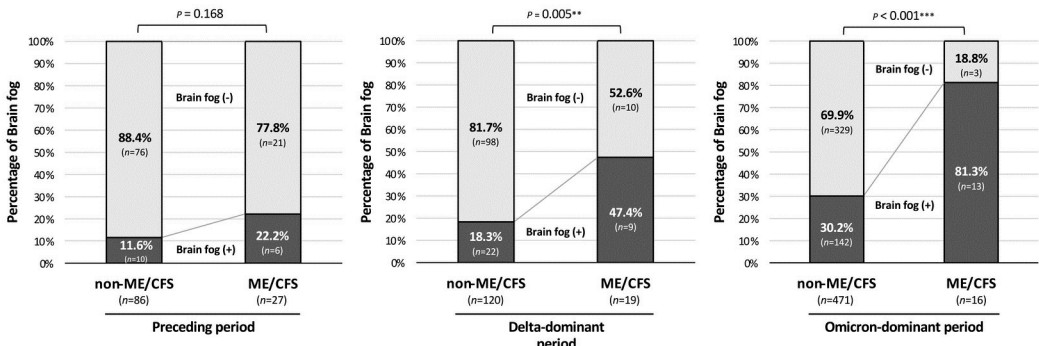

**Fig 4. Prevalence of brain fog symptoms in long COVID patients in the ME/CFS group and non-ME/CFS group.** Percentages of patients who had brain fog symptoms are shown on the basis of the onset periods of COVID-19 (Total: n = 739; Preceding period: n = 113; Delta-dominant period: n = 139; Omicron-dominant period: n = 487). The $\chi^2$ test was performed for the proportions of patients with brain fog between the two groups in each variant phase, and **$p$<0.01 and ***$p$<0.001 were considered statistically significant.

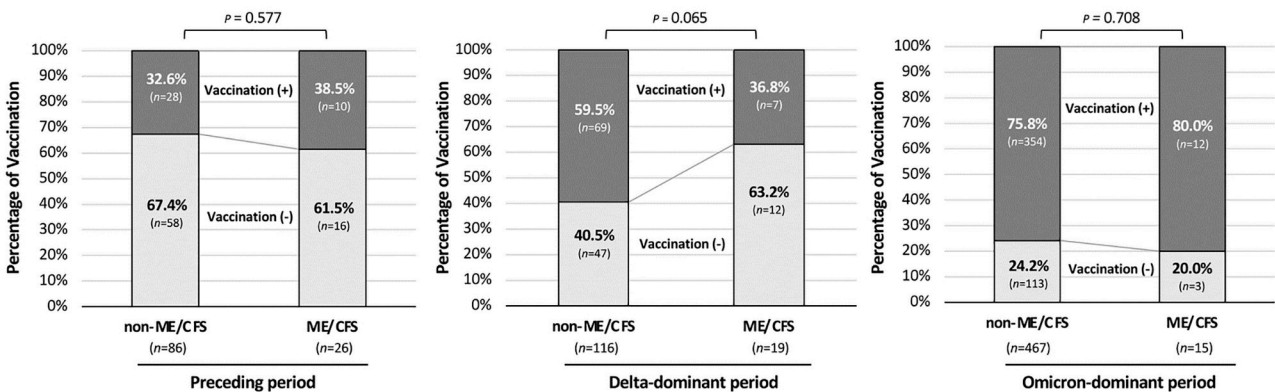

**Fig 5. Vaccination rates in the ME/CFS group and non-ME/CFS group.** Percentages of vaccinated patients are shown on the basis of the onset periods of COVID-19 (Total: n = 729; Preceding period: n = 112; Delta-dominant period: n = 135; Omicron-dominant period: n = 482). The $\chi^2$ test was performed for the proportions of vaccinated patients between the two groups in each variant phase.

Throughout the entire study period, the ME/CFS group had a significantly higher proportion of unvaccinated patients as mentioned above (Table 3), whereas there was generally no significant difference in the vaccination rates between the ME/CFS group and the non-ME/CFS group in each viral-variant period as shown in Fig 5. In the Preceding period, the vaccination rate was low at approximately 34% (38 of 112 cases) and there was no significant association between the vaccination status and the prevalence of long COVID-related ME/CFS. In the Delta-dominant period with a vaccination rate of 56% (76 of 135 cases), it was revealed that patients who had never been vaccinated had a trend toward an increasing rate of transition to ME/CFS ($p = 0.065$). In the Omicron-dominant period, when the whole vaccination rate increased up to 76% (366 of 482 cases), no significant association was observed between vaccination status and the transition to ME/CFS.

## Discussion

Analysis of data for 739 patients who visited our specialized clinic clarified the clinical characteristics of long COVID patients in whom ME/CFS may develop and showed that the prevalence of ME/CFS in long COVID patients was 8.4%. ME/CFS was diagnosed in the long COVID patients after face-to-face consultation and after excluding the possibility of other diseases. Moreover, to the best of our knowledge, the present study is the first study to investigate the differences in the prevalence of ME/CFS in long COVID patients who had been infected with different viral variants. The long COVID patients who had been infected in the Omicron phase had a significantly lower prevalence of ME/CFS, which was reduced by 86% from the Preceding phase and by 43% from the Delta phase. On the other hand, ME/CFS patients who had COVID-19 during the Omicron phase had a significantly higher incidence of brain fog. The results suggest that there is a difference regarding the pathophysiology of long COVID depending on the viral variants.

ME/CFS is known as a post-infectious fatigue syndrome [27] and the manifestation of persistent fatigue following COVID-19 has been recognized as a condition of ME/CFS related to long COVID [28]. However, there have been few studies showing an association between ME/CFS and severity of the initial illness [8, 29]. There was also a lack of data for Omicron COVID-19 patients in previous studies. The present study showed a descending prevalence of ME/CFS with COVID-19 from the Preceding period to the Delta-dominant period and to the Omicron-dominant period. In other words, the frequency of the development of ME/CFS in

long COVID patients decreased with change in the dominant viral strain. This might be due to differences in the severity of COVID-19 in the acute phase among variants or the development of resistance to exacerbation due to the prolonged COVID-19 pandemic that led to widespread vaccination and acquired immunity. There are conflicting opinions regarding the association between the severity of COVID-19 in the acute period and ME/CFS induced by COVID-19. However, the present study showed that the severity of COVID-19 in the acute phase was significantly greater in the ME/CFS group than in the non-ME/CFS group.

In this study, it was revealed that the ME/CFS group had a significantly lower rate of vaccination than that in the non-ME/CFS group over the study period. Interestingly, a comparison in each variant period showed that there was no significant difference in vaccination rate between ME/CFS patients and non-ME/CFS patients in any period. There was a tendency for the development of ME/CFS in long COVID patients without vaccination compared to that in long COVID patients who were vaccinated only in the Delta-dominant period ($p = 0.065$). On the other hand, while the vaccination rate increased for both ME/CFS and non-ME/CFS patients, the vaccination rate had no direct association with transition to ME/CFS in the Omicron-dominant period. In this regard, it was reported that COVID-19 patients in the Omicron era were less likely to develop long COVID than were COVID-19 patients in the pre-Delta and Delta eras combined due to vaccination and the viral era-related effect [30]. Considering these findings, it is thought that long COVID and ME/CFS share a common trigger among factors that a COVID-19 vaccine can mitigate such as severity in the acute phase of COVID-19. In addition, it is possible that, particularly in the case of Omicron variants, the transition from long COVID to ME/CFS is driven by a trigger that differs from the acute-phase severity and is difficult to prevent by vaccination.

Another possible reason for the lower prevalence of ME/CFS in the Omicron-dominant period in our study is the lower proportion of long COVID patients meeting the Canadian Consensus Criteria [11, 13]. The Canadian Criteria are considered to be useful for identifying relatively severe cases of ME/CFS because the criteria include not only fatigue, sleep disturbances and cognitive impairments but also dysfunctions of the autonomic nervous system (ANS), neuroendocrine system and immune system [11]. The criteria have diagnostic items for identifying patients who have relatively severe symptoms of ME/CFS [11], and the results of the present study thus indicate that there was a relatively small number of long COVID patients infected with Omicron variants in whom moderate to severe ME/CFS conditions involving systemic dysfunctions developed.

In the present study, serum ferritin levels were apparently elevated in female patients in the ME/CFS group who had COVID-19 in the Preceding period. These results for serum ferritin levels suggest that severe inflammation in the acute phase of COVID-19 is related to ME/CFS with long COVID. We showed in a previous study that serum ferritin level is a possible predictive factor for ME/CFS with long COVID [12], especially in female patients, and serum ferritin levels were also reported to be decreased with improvement in symptoms of ME/CFS [26]. Significant differences in serum ferritin levels between ME/CFS patients and non-ME/CFS patients were not found except for female long COVID patients who were infected in the Preceding phase, which may imply that these patients might have obtained some anti-inflammatory activity to mitigate the risk of developing ME/CFS such as anti-inflammatory activity from vaccination or acquired immunity during the Delta- and Omicron-dominant phases. It was of interest that the results for serum ferritin levels were consistent with the decreasing trend in the occurrence of ME/CFS as a transition from long COVID.

SDS scores for healthy individuals range from 20 to 40 [31]. On the other hand, in long COVID patients, even patients who did not develop ME/CFS had a higher SDS score of 48 as a median. Also, in long COVID patients with ME/CFS, the median SDS score was 51, showing a

moderate increase. Since the SDS score was obtained at the first visit, it is possible that ME/CFS patients had been in a depressed condition from the early stages. However, it is presumed that the higher SDS score in ME/CFS patients was influenced by the severe physical fatigue experienced by the patients, given that SDS is a self-administered questionnaire.

In the present study, fatigue and headache were commonly frequent symptoms of the patients with ME/CFS related to long COVID in each variant period, being consistent with the clinical characteristics of ME/CFS. It has been reported that long COVID patients who were infected with Omicron variants had lower rates of dysgeusia, dysosmia and hair removal [19], and the frequencies of these symptoms were also decreased in long COVID patients with or without ME/CFS during the Omicron-dominant period in the present study. Therefore, it can be reaffirmed that ME/CFS related to COVID-19 has the aspect of a condition that transitioned from long COVID. While poor concentration was not a characteristic symptom in long COVID patients infected during the Omicron-dominant period, our study suggested that poor concentration is a representative symptom in ME/CFS patients in the Omicron-dominant period.

It was also shown that patients with ME/CFS who had been infected with Omicron variants had more frequent complaints of headache, insomnia, poor concentration, and memory disorder than did patients with ME/CFS caused by other variants. These symptoms that were more frequent in ME/CFS patients infected with Omicron variants are explainable by virus-induced neurological dysfunction [32], and the results of the present study suggest that Omicron variants have a higher potential to damage the central nervous system (CNS) and that the development of ME/CFS is a result of this potential being strongly manifested. Certainly, COVID-19 has been shown to damage the CNS in various ways. For example, the virus can pass into the bloodstream through alveolar epithelial cells having the angiotensin-converting enzyme 2 (ACE2) receptor and then invade the CNS by damaging endothelial cells of the blood-brain barrier (BBB) or circumventing the BBB through periventricular organs or the choroid plexus [33, 34]. The virus intrudes on the brain by infecting monocytes/macrophages or passing along peripheral nerves such as the olfactory nerve [35–37]. There is a possibility that the types of nerves invaded and the mechanisms by which the CNS is damaged vary depending on the viral strains, which could give rise to differences in the clinical pictures caused by the variants of the virus.

ME/CFS patients have been reported to suffer cognitive impairments such as lack of memory and attention, difficulty in word retrieval, and thought difficulties, which are sometimes referred to as brain fog [38, 39]. On the other hand, in long COVID patients with brain fog, an increase in serum inflammatory markers has been observed [40] and the severity of the acute phase is considered to be linked to the transition to ME/CFS. In the present study, despite the significant decline in the number of patients with ME/CFS related to long COVID, the prevalence of brain fog in the long COVID patients in whom ME/CFS developed was increased in the Omicron-dominant phase with a 3.6-fold increase in prevalence from that in the Preceding period and a 1.7-fold increase in prevalence from that in the Delta period. In this regard, damage to the BBB has been suggested in long COVID patients with brain fog [40], and the Omicron strain is thought to be more likely than the other strains to cause destruction of BBB-cellular components [41]. These findings suggest that the differences in neural tropism among the virus strains and/or the prolonged duration of the COVID-19 pandemic might have caused the increase in the occurrence of brain fog, rather than the severity of infection and inflammatory response of COVID-19. The higher prevalence of brain fog, particularly in ME/CFS cases with long COVID from infection during the Omicron phase, may be attributed to the variant-specific characteristics.

This retrospective study has several limitations. First, the present study was conducted to investigate the proportion and characteristics of long COVID-19 patients with ME/CFS who visited an outpatient clinic of a single center in Japan. Second, we may not have completely included all of the ME/CFS patients since the follow-up periods were limited at most to three years. Third, the study included only patients who were referred to a specific outpatient clinic, and the entire population of long COVID patients was not investigated. Fourth, we assumed viral variants based on the phase of infection instead of detailed genetic analysis. Moreover, we could not completely eliminate the possibility of repeated onsets of COVID-19. Fifth, we investigated the association between the presence or absence of vaccination at the first visit and the prevalence of ME/CFS, but we could not obtain information on other factors such as the number of vaccinations or the correlation between the COVID-19 onset date and the vaccination date. Finally, the number of patients infected in the Omicron phase was much larger than the numbers of patients infected in other phases in this study, and that difference might have influenced the statistics for comparing the viral variants. Despite these limitations and situations, the present study provided the first results regarding the real clinical pictures of the phase-dependent differences in long COVID including ME/CFS and the results are valuable for characterizing patients with long COVID due to Omicron variants, which are expected to continue increasing in number.

Collectively, the results of the present study revealed that the overall prevalence of ME/CFS in long COVID patients was 8.4%. Our study was a retrospective study and utilized a stricter diagnostic method that required meeting three sets of criteria for ME/CFS. Thus, the prevalence of ME/CFS in this study might have been underestimated compared to that in other studies in which only one international set of criteria for ME/CFS was used. However, to the best of our knowledge, this study is the first study showing a variant-dependent reduction in the prevalence of ME/CFS related to long COVID and an increase in the rate of brain fog that is inversely correlated with the prevalence of ME/CFS. In addition, this study suggested a preventive effect of a COVID-19 vaccine against ME/CFS and unique pathogenicity of the Omicron variants. Future subgroup analyses of Omicron subvariants and vaccination factors such as the number of doses and duration between the date of vaccination and COVID-19 onset are needed.

## Acknowledgments

We are sincerely grateful to the clinical and office staff at the Department of General Medicine who contributed to the present work.

## Author Contributions

**Conceptualization:** Fumio Otsuka.

**Data curation:** Yuki Otsuka, Hiroyuki Honda, Yasuhiro Nakano, Naruhiko Sunada, Yasue Sakurada, Yui Matsuda, Yoshiaki Soejima, Keigo Ueda.

**Formal analysis:** Satoru Morita, Kazuki Tokumasu, Yuki Otsuka, Hiroyuki Honda, Yasuhiro Nakano, Naruhiko Sunada, Yasue Sakurada, Yui Matsuda.

**Funding acquisition:** Fumio Otsuka.

**Investigation:** Satoru Morita, Kazuki Tokumasu, Yuki Otsuka, Hiroyuki Honda, Yasuhiro Nakano, Naruhiko Sunada, Yoshiaki Soejima.

**Methodology:** Kazuki Tokumasu, Hiroyuki Honda.

**Supervision:** Keigo Ueda, Fumio Otsuka.

**Validation:** Kazuki Tokumasu, Yuki Otsuka, Yasuhiro Nakano, Yasue Sakurada, Yui Matsuda, Yoshiaki Soejima, Keigo Ueda.

**Writing – original draft:** Satoru Morita, Kazuki Tokumasu, Fumio Otsuka.

**Writing – review & editing:** Keigo Ueda.

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
