## [Decision Letter · Decision Letter 0]

19 Aug 2024

PONE-D-24-13000Phase-dependent trends in the prevalence of myalgic encephalomyelitis / chronic fatigue syndrome (ME/CFS) related to long COVID in Japan.PLOS ONE

Dear Dr. Otsuka,

Thank you for submitting your manuscript to PLOS ONE. After careful consideration, we feel that it has merit but does not fully meet PLOS ONE’s publication criteria as it currently stands. Therefore, we invite you to submit a revised version of the manuscript that addresses the points raised during the review process.

Please note that we have only been able to secure a single reviewer to assess your manuscript. We are issuing a decision on your manuscript at this point to prevent further delays in the evaluation of your manuscript. Please be aware that the editor who handles your revised manuscript might find it necessary to invite additional reviewers to assess this work once the revised manuscript is submitted. However, we will aim to proceed on the basis of this single review if possible. Comments from PLOS Editorial Office: We note that the reviewers has recommended that you cite specific previously published works. As always, we recommend that you please review and evaluate the requested works to determine whether they are relevant and should be cited. It is not a requirement to cite these works. We appreciate your attention to this request.

We look forward to receiving your revised manuscript.

Kind regards,

Annesha Sil, Ph.D.

Associate Editor

PLOS ONE

“This research was supported by AMED under Grant Number (22fk0108517h0001).”

“None”

Reviewers' comments:

Reviewer's Responses to Questions

**Comments to the Author**

1. Is the manuscript technically sound, and do the data support the conclusions?

Reviewer #1: Partly

2. Has the statistical analysis been performed appropriately and rigorously? 

Reviewer #1: Yes

3. Have the authors made all data underlying the findings in their manuscript fully available?

Reviewer #1: Yes

4. Is the manuscript presented in an intelligible fashion and written in standard English?

Reviewer #1: Yes

5. Review Comments to the Author

Reviewer #1: Page 4, line 1 - General fatigue alone is not considered a diagnostic criteria for ME/CFS, it should specifically be associated with substantial reduction in function and not alleviated by rest per Fukuda and IOM criteria. "General fatigue" is misleading in its simplicity and risks misdiagnosis.

Page 4, line 10 - Multiple studies estimate 50% of Long COVID patients have ME/CFS: 10.3389/fneur.2023.1090747, 10.1016/j.eclinm.2023.102146, 10.3390/neurolint15010001 as some examples.

Page 7, line 9-12 - Applied how? If they met any of the criteria they could be considered ME/CFS? Good to clarify this for readers. (*addendum: now seeing under results that they needed to meet all three; the ME/CFS experts I know do not do this so this seems very concerning for underestimating prevalence)

Page 7, line 14 - Not sure what this was saying - so if there was a possibility of another condition, CCC was used to confirm if they had it? ME/CFS is not a diagnosis of exclusion, and you are using CCC to decide whether or not to include them which is also the strictest criteria (many of our own LC-ME do not meet the criteria because of the pain requirement). This may be affecting your data and may explain the discrepancy between yours and higher rates of ME/CFS and LC report. Make sure to comment on this in your discussion, especially as far as how "having conditions other than ME/CFS" was assessed (methods as well). (*addendum: now that I'm seeing all three are applied, why is this line needed at all?)

Page 8, line 6-8 - Confused about this as it was a retrospective study, was there a greater study in your clinic that they were made aware of and able to opt out of? Or was it actually posted on a website that you'd be doing a chart review and that people could opt out? If the latter, may want to include that earlier in the methods because the order is currently confusing.

Page 9, line 17-18 through page 10, line 1-2 - Were the habits pre/post-COVID? I would be very surprised if this something that was kept up after COVID as there are replicated studies in ME/CFS that a majority are unable to tolerate alcohol, which is our clinical experience with ME/CFS and LC-ME as well: https://doi.org/10.3389%2Ffped.2019.00012, https://doi.org/10.1016%2FS0022-3999%2803%2900077-1, https://doi.org/10.1046%2Fj.1365-2796.2001.00890.x

Page 10, line 5-9 - Within the same epochs, or of ME/CFS patients in general? ME/CFS was more common after initial COVID waves when there was no vaccination available, important to assess by subgroup to see if it was variant effect or true vaccination effect.

Page 17, line 8-14 - If you are going to discuss the association depression and LC-ME, please be sure to discuss that it is very appropriate for a person with new debilitating disease to experience depression, especially when quality of life in ME/CFS has been rated worse than cancers, COPD, renal disease, etc. There is a difference between treating co-existing mood disorders as a result of chronic illness compared to implementing depression treatment as management for LC/ME/CFS. Mood tends to normalize to general population levels after about 2 years. I would recommend therefore discussing depression management as an adjunct. Additionally, depression screens are often flawed as the symptoms associated with ME/CFS may be used in depression screens (for example, PHQ will automatically be elevated even in a non-depressed ME/CFS patient because of fatigue, moving less, eating less even though those are not related to depression in that patient). Be sure to discuss this confounding factor if present.

Overall, the study addresses an important question of prevalance of ME/CFS based on variant, with results correlating with the experience of our Long COVID clinics as well. However, I question the methodology for diagnosis for the study. The determination of who was included as ME/CFS is very limiting, as I have not known any other clinical ME/CFS expert to only diagnose based on meeting IOM, Fukuda, and CCC together (why Fukuda from 1994 which was replaced by the IOM 2015, and not ICC for a third?), and it is not a diagnosis of exclusion. I suspect this is why the number of 8.4% was reached in contrast to studies estimating 43-58%. Additionally, it is unclear to me if the authors thought carefully about confounding factors related to things like vaccination (was vaccination lower in ME/CFS because of the timing of the first variant, or was it still more common in later variants when vaccines were available too?) and depression questionnaire overlaps with symptoms. It seems to me that the study would either benefit from the authors reconsidering the way the data is reported, or being very thorough in the discussion to explain potential pitfalls.

6. PLOS authors have the option to publish the peer review history of their article (what does this mean?). If published, this will include your full peer review and any attached files.

Reviewer #1: No

---

## [Author Response · Author response to Decision Letter 0]

3 Oct 2024

Responses to the Referee’s comments:

Comment: Page 4, line 1 - General fatigue alone is not considered a diagnostic criteria for ME/CFS, it should specifically be associated with substantial reduction in function and not alleviated by rest per Fukuda and IOM criteria. "General fatigue" is misleading in its simplicity and risks misdiagnosis.

Response: As the referee suggested, fatigue in ME/CFS patients is not merely systemic fatigue but rather than new onset of unexplained physical and mental fatigue accompanied by significant functional impairment. To clearly distinguish fatigue of ME/CFS from common fatigue, we have amended the term “general fatigue” to “pathological fatigue” and added a sentence explaining the details of fatigue in ME/CFS following the term “pathological fatigue”. In addition, to characterize the “pathological fatigue” in long COVID patients, we have further analyzed the complicated symptoms with fatigue in the ME/CFS group and the non-ME/CFS group. In the first manuscript, we stated that the ME/CFS group had a significantly higher rate of fatigue than that in the non-ME/CFS group in the Omicron-dominant period as shown in Fig. 2. In the revised manuscript, we added a comparison of the differences in the prevalences of major symptoms between the ME/CFS and non-ME/CFS groups during the three variant periods, showing that fatigue and headache were two major and common symptoms of ME/CFS related to long COVID regardless of the variant period (as revised Fig. 2). Based on these new data, we consider that pathological fatigue can be more clearly characterized in patients with ME/CFS related to long COVID. Thank you for the important comments.

Comment: Page 4, line 10 - Multiple studies estimate 50% of Long COVID patients have ME/CFS: 10.3389/fneur.2023.1090747, 10.1016/j.eclinm.2023.102146, 10.3390/neurolint15010001 as some examples.

Response: We appreciate the opportunity to clarify this point. Certainly, there are some reports showing a much higher possibility of ME/CFS related to long COVID. In the introduction section of our revised manuscript, the suggested papers were cited and the data were presented. The prevalence of ME/CFS in patients with long COVID may have a wide range, which we consider results from differences in the methods for diagnosis of ME/CFS and, in part, the timing of COVID-19 onset among the patients included in the study. Firstly, biomarkers specific to ME/CFS have not been identified yet and diagnostic criteria for ME/CFS are not unified, which may be involved in the variability of the prevalence among studies. Secondly, it has been revealed that the temporal change during the COVID-19 pandemic and the widespread distribution of vaccination have contributed to changes in the transition rate to ME/CFS. This suggests that evaluating the rate for each viral variant is more clinically relevant. Therefore, the lower prevalence in this study is not an error but is due to more careful diagnosis and a broader patient inclusion period. In fact, the present study showed that the prevalence of ME/CFS changed with transition of the COVID-19 pandemic. We addressed this issue in the introduction section of the revised manuscript.

Comment: Page 7, line 9-12 - Applied how? If they met any of the criteria they could be considered ME/CFS? Good to clarify this for readers. (*addendum: now seeing under results that they needed to meet all three; the ME/CFS experts I know do not do this so this seems very concerning for underestimating prevalence)

Response: Thank you for your question. As you point out, in our study, medical records were examined retrospectively to determine whether the diagnostic criteria were met or not. Therefore, the prevalence of ME/CFS might be underestimated. This is also a limitation of the study and we therefore added this issue to the limitations of the study. In addition, the method for diagnosing ME/CFS by meeting all three sets of diagnostic criteria is unique to our study. However, we think this method did not contribute to underestimation of the prevalence of ME/CFS. As shown in Table 3, it can be observed in the Preceding period and the Delta-dominant period that the prevalence of ME/CFS remains nearly the same whether evaluated by a single set of diagnostic criteria or all three sets of criteria. The prevalence of ME/CFS in the Omicron-dominant period was lower than that in the other periods, and the lower prevalence is due to the very low percentage of patients who met CCC and not because of the use of all three sets of diagnostic criteria. As described in the limitations section, the possibility that the large number of patients with COVID-19 during the Omicron-dominant period contributes to the underestimation of the prevalence of ME/CFS cannot be ruled out. However, the Omicron variant has spread the most widely and has the largest number of infections, and we consider it is natural to include a large number of patients infected during the Omicron-dominant period. We discussed this issue in the revised manuscript.

Comment: Page 7, line 14 - Not sure what this was saying - so if there was a possibility of another condition, CCC was used to confirm if they had it? ME/CFS is not a diagnosis of exclusion, and you are using CCC to decide whether or not to include them which is also the strictest criteria (many of our own LC-ME do not meet the criteria because of the pain requirement). This may be affecting your data and may explain the discrepancy between yours and higher rates of ME/CFS and LC report. Make sure to comment on this in your discussion, especially as far as how "having conditions other than ME/CFS" was assessed (methods as well). (*addendum: now that I'm seeing all three are applied, why is this line needed at all?)

Response: We appreciate the opportunity to clarify this point. We agree with the referee’s points indicating that the diagnosis is not simply an exclusion diagnosis. We amended and added a detailed explanation of the diagnosis of ME/CFS related to long COVID. The diagnosis of ME/CFS is made by meeting criteria such as CCC, but it is important to note that other diseases can also meet these diagnostic criteria. CCC include examples of diseases that need to be differentiated and conditions that can coexist with ME/CFS, which are referred to when diagnosing ME/CFS. Face-to-face examinations were conducted by physicians for all patients and, when needed, hormones and autoantibodies were measured, imaging tests and orthostatic tests were performed, and patients were referred to psychiatrists. By carefully evaluating the possibility of other conditions in this manner, we retrospectively diagnosed ME/CFS using all three sets of diagnostic criteria based on medical records. We addressed this issue in the methods section of the revised manuscript.

Comment: Page 8, line 6-8 - Confused about this as it was a retrospective study, was there a greater study in your clinic that they were made aware of and able to opt out of? Or was it actually posted on a website that you'd be doing a chart review and that people could opt out? If the latter, may want to include that earlier in the methods because the order is currently confusing.

Response: Thank you for your inquiry. The study was a retrospective observational study and we did not obtain any prospective data or implement any new interventions. Therefore, informed consent was not required to be obtained from patients, and the study was published on the hospital website and patients themselves were given the opportunity to not have personal data used for the study. This is an ethical consideration, and the Ethics Board has approved this study. We addressed this issue in the revised manuscript.

Comment: Page 9, line 17-18 through page 10, line 1-2 - Were the habits pre/post-COVID? I would be very surprised if this something that was kept up after COVID as there are replicated studies in ME/CFS that a majority are unable to tolerate alcohol, which is our clinical experience with ME/CFS and LC-ME as well: https://doi.org/10.3389%2Ffped.2019.00012, https://doi.org/10.1016%2FS0022-3999%2803%2900077-1, https://doi.org/10.1046%2Fj.1365-2796.2001.00890.x

Response: Thank you for the comment. However, as described in the Patients and Methods (page 6, line 14), we retrospectively collected clinical information from medical records for our long COVID patients. Thus, understanding whether habits such as alcohol drinking were pre- or post-COVID is considered a limitation of retrospective studies. We added this issue in the limitations section of the revised manuscript.

Comment: Page 10, line 5-9 - Within the same epochs, or of ME/CFS patients in general? ME/CFS was more common after initial COVID waves when there was no vaccination available, important to assess by subgroup to see if it was variant effect or true vaccination effect.

Response: We thank the reviewer for the careful review of the manuscript. We investigated and the proportions of vaccinated patients with long COVID in the ME/CFS group and the non-ME/CFS group in different periods (as shown in revised Fig. 5). In fact, there was generally no significant difference in vaccination rates between the ME/CFS group and the non-ME/CFS group in each viral variant period. However, our additional data may imply that vaccination has a possible preventive effect on the transition to ME/CFS. From the Preceding period to the Omicron period, patients without vaccination were more like to develop ME/CFS than were patients with vaccination, suggesting that vaccination might reduce the risk of a severe condition in the acute phase of COVID-19 and lower the possibility of transitioning to ME/CFS. However, other factors such as the number of vaccine doses and the interval between the vaccination date and onset of COVID-19 that might affect the transition rate to ME/CFS were not clear in the present study and further research is needed. We added these points in the limitation sections of the revised discussion.

Comment: Page 17, line 8-14 - If you are going to discuss the association depression and LC-ME, please be sure to discuss that it is very appropriate for a person with new debilitating disease to experience depression, especially when quality of life in ME/CFS has been rated worse than cancers, COPD, renal disease, etc. There is a difference between treating co-existing mood disorders as a result of chronic illness compared to implementing depression treatment as management for LC/ME/CFS. Mood tends to normalize to general population levels after about 2 years. I would recommend therefore discussing depression management as an adjunct. Additionally, depression screens are often flawed as the symptoms associated with ME/CFS may be used in depression screens (for example, PHQ will automatically be elevated even in a non-depressed ME/CFS patient because of fatigue, moving less, eating less even though those are not related to depression in that patient). Be sure to discuss this confounding factor if present.

Response: We appreciate the reviewer’s suggestions. We certainly consider that ME/CFS is not a psychologic status of depression. In this paragraph in the original manuscript, our intention was simply to highlight that the SDS score tended to be higher in the ME/CFS group and to emphasize the necessity of thoroughly considering the possibility that psychiatric or neurological disorders, such as depression, may be masked when diagnosing ME/CFS. As the reviewer mentioned, we also agree that the association between depression and ME/CFS requires more careful discussion and further research. In the revised manuscript, we discussed the reason for the ME/CFS group having a high SDS score. In fact, patients in the ME/CFS group already had a high SDS score at the first visit, suggesting that ME/CFS patients had been in a depressed condition from an early stage. However, it was also possible that their severe physical fatigue or PEM might make the SDS score higher than actual because SDS is an auto-administered questionnaire. We added this discussion in the revised manuscript.

Comment: Overall, the study addresses an important question of prevalence of ME/CFS based on variant, with results correlating with the experience of our Long COVID clinics as well. However, I question the methodology for diagnosis for the study. The determination of who was included as ME/CFS is very limiting, as I have not known any other clinical ME/CFS expert to only diagnose based on meeting IOM, Fukuda, and CCC together (why Fukuda from 1994 which was replaced by the IOM 2015, and not ICC for a third?), and it is not a diagnosis of exclusion. I suspect this is why the number of 8.4% was reached in contrast to studies estimating 43-58%. Additionally, it is unclear to me if the authors thought carefully about confounding factors related to things like vaccination (was vaccination lower in ME/CFS because of the timing of the first variant, or was it still more common in later variants when vaccines were available too?) and depression questionnaire overlaps with symptoms. It seems to me that the study would either benefit from the authors reconsidering the way the data is reported, or being very thorough in the discussion to explain potential pitfalls.

Response: Thank you for recognizing the importance and the value of our research. Regarding the method for ME/CFS diagnosis, we aimed to establish the clinical picture of ME/CFS patients. Due to a lack of specific biomarkers for ME/CFS, there are approximately 20 sets of diagnostic criteria based on clinical features. Each set of criteria defines ME/CFS differently, reflecting distinct clinical presentations based on the researchers’ unique clinical experience, which may cause the difference in the prevalence of ME/CFS among sets of criteria. Therefore, we consider that using multiple sets of diagnostic criteria together enables a more accurate diagnosis of ME/CFS. The reason we chose Fukuda, CCC, and IOM criteria is that these sets of criteria have been the most frequently utilized criteria in the world. In fact, our diagnostic method that requires meeting all three sets of criteria was unique and was so strict that it might have led to underestimating the prevalence of ME/CFS. However, as shown in Table 2, there was little difference between the prevalences of ME/CFS based on a single set of criteria and based on all three sets of criteria at least in the Preceding and Delta-dominant periods. Moreover, we have a preceding study for ME/CFS related to long COVID in which the same three sets of diagnostic criteria were used, and it was concluded that the prevalence of ME/CFS in our study was close to the prevalence reported previously. Therefore, we believe that our unique method for diagnosis of ME/CFS is less likely to underestimate the prevalence than expected and is rather close to the real-world data. As for the association between vaccination and the prevalence of ME/CFS, it was found that ME/CFS patients were significantly less likely to be vaccinated over the entire period. On the other hand, a comparison in each variant period showed that there was no significant difference in the vaccination rate between the ME/CFS group and the non-ME/CFS group in any period. However, in the Delta-dominant period, long COVID patients without vaccination tended to develop ME/CFS compared to long COVID patients with vaccination. Although further research such as subgroup analysis is needed to clarify the association between COVID-19 vaccine and transition to ME/CFS from long COVID, this study suggests that vaccination can prevent not only the onset and aggravation of COVID-19 but also the development of ME/CFS due to COVID-19. As for depression, we did not intend to discuss the relationship with depression and ME/CFS in detail. In the original manuscript, we wanted to show the importance of excluding depression because the ME/CFS group had a high SDS score and depression is one of the conditions to rule out for diagnosing ME/CFS. Nonetheless, we added discussion about neuropsychiatric conditions regarding long COVID, which might mask the inherent purpose of the paragraph. In the 

---

## [Decision Letter · Decision Letter 1]

26 Nov 2024

Phase-dependent trends in the prevalence of myalgic encephalomyelitis / chronic fatigue syndrome (ME/CFS) related to long COVID: A criteria-based retrospective study in Japan.

PONE-D-24-13000R1

Dear Dr. Otsuka,

We’re pleased to inform you that your manuscript has been judged scientifically suitable for publication and will be formally accepted for publication once it meets all outstanding technical requirements.

Kind regards,

Etsuro Ito, Ph.D.

Academic Editor

PLOS ONE

Reviewers' comments:

Reviewer's Responses to Questions

**Comments to the Author**

1. If the authors have adequately addressed your comments raised in a previous round of review and you feel that this manuscript is now acceptable for publication, you may indicate that here to bypass the “Comments to the Author” section, enter your conflict of interest statement in the “Confidential to Editor” section, and submit your "Accept" recommendation.

Reviewer #1: All comments have been addressed

2. Is the manuscript technically sound, and do the data support the conclusions?

Reviewer #1: Yes

3. Has the statistical analysis been performed appropriately and rigorously? 

Reviewer #1: Yes

4. Have the authors made all data underlying the findings in their manuscript fully available?

Reviewer #1: Yes

5. Is the manuscript presented in an intelligible fashion and written in standard English?

Reviewer #1: Yes

6. Review Comments to the Author

Reviewer #1: Thank you for being so thorough in your reassessment and revisions of the manuscript. Best wishes with your publication!

7. PLOS authors have the option to publish the peer review history of their article (what does this mean?). If published, this will include your full peer review and any attached files.

Reviewer #1: No

---

## [Editor Report · Acceptance letter]

28 Nov 2024

PONE-D-24-13000R1 

PLOS ONE

Dear Dr. Otsuka, 

I'm pleased to inform you that your manuscript has been deemed suitable for publication in PLOS ONE. Congratulations! Your manuscript is now being handed over to our production team.

Kind regards, 

on behalf of

Prof. Etsuro Ito 

Academic Editor

PLOS ONE